# A new model for predicting the outcome and effectiveness of drug therapy in patients with severe fever with thrombocytopenia syndrome: A multicenter Chinese study

**Guomei Xia[1], Shanshan Sun[1], Shijun Zhou[1], Lei Li[2], Xu Li[3], Guizhou Zou[1], Cheng Huang[4], Jun Li[4], Zhenhua Zhang** 🄳 [1,4]*

**1** Institute of Clinical Virology, Department of Infectious Diseases, The Second Hospital of Anhui Medical University, Hefei, China, **2** Department of Infectious Diseases, Anhui Provincial Hospital of Anhui Medical University, Hefei, China, **3** Department of Infectious Diseases, The First Affiliated Hospital of Anhui Medical University, Hefei, China, **4** Inflammation and Immune Mediated Diseases Laboratory of Anhui Province, Anhui Institute of Innovative Drugs, School of Pharmacy, Anhui Medical University, Hefei, China

* zzh1974cn@163.com

**Data Availability Statement:** All relevant data are within the manuscript and it's Supporting Information files.

## Abstract

### Background

There are a few models for predicting the outcomes of patients with severe fever with thrombocytopenia syndrome (SFTS) based on single-center data, but clinicians need more reliable models based on multicenter data to predict the clinical outcomes and effectiveness of drug therapy.

### Methodology/principal findings

This retrospective multicenter study analyzed data from 377 patients with SFTS, including a modeling group and a validation group. In the modeling group, the presence of neurologic symptoms was a strong predictor of mortality (odds ratio: 168). Based on neurologic symptoms and the joint indices score, which included age, gastrointestinal bleeding, and the SFTS virus viral load, patients were divided into double-positive, single-positive, and double-negative groups, which had mortality rates of 79.3%, 6.8%, and 0%, respectively. Validation using data on 216 cases from two other hospitals yielded similar results. A subgroup analysis revealed that ribavirin had a significant effect on mortality in the single-positive group (P = 0.006), but not in the double-positive or double-negative group. In the single-positive group, prompt antibiotic use was associated with reduced mortality (7.2% vs 47.4%, P < 0.001), even in individuals without significant granulocytopenia and infection, and early prophylaxis was associated with reduced mortality (9.0% vs. 22.8%, P = 0.008). The infected group included SFTS patients with pneumonia or sepsis, while the noninfected group included patients with no signs of infection. The white blood cell count and levels of C-reactive protein and procalcitonin differed significantly between the infection and non-infection groups (P = 0.020, P = 0.011, and P = 0.003, respectively), although the absolute difference in the medians were small.

**Funding:** The study was supported by Anhui Provincial Natural Science Foundation (grant number 2108085MH298 to ZZ) and the Scientific research project of Anhui Medical University (grant number 2019GMFY02 and 2021lcxk027 both to ZZ). The funders had no role in the study design, data collection and analysis, decision to publish, or preparation of the manuscript.

**Competing interests:** None

## Conclusions/Significance

We developed a simple model to predict mortality in patients with SFTS. Our model may help to evaluate the effectiveness of drugs in these patients. In patients with severe SFTS, ribavirin and antibiotics may reduce mortality.

---

### Author summary

Severe fever with thrombocytopenia syndrome (SFTS) was first reported in 2011 and 11.2–30% of patients die of multiple organ failure. Previous prognostic models for SFTS are mostly based on single-center data and lack validation. Moreover, the drug efficacy of ribavirin and gamma globulin is controversial. Here, we developed a simple model for predicting the mortality and help to evaluate the effectiveness of drugs in patients with SFTS. In this study, the data of three hospitals were used to establish and verify the model. Based on neurologic symptoms and the joint indices score, which considered age, gastrointestinal bleeding, and SFTSV RNA, patients were divided into double-positive, single-positive, and double-negative groups, which had mortality rates of 79.3%, 6.8%, and 0% in the modeling groups, respectively. Validation yielded similar results. Subgroup analysis revealed that ribavirin had a significant effect on mortality in the single-positive group but not in the other two groups. In the single-positive group, prompt antibiotic use was associated with reduced mortality, even in individuals without significant granulocytopenia and infection, and early prophylaxis was associated with reduced mortality.

## Introduction

Severe fever with thrombocytopenia syndrome (SFTS), which is caused by the SFTS virus (SFTSV), is an emerging infectious disease that was first identified in rural areas of China in 2011 [1]. SFTS is characterized by fever, fatigue, thrombocytopenia, leukopenia, and gastrointestinal and central nervous system symptoms, and it severely affects patients, who often die of multiple organ failure (mortality rate: 11.2–30%) [2–5]. The spread of SFTSV has become an important public health issue in China [6]. SFTS has been detected in at least 20 Chinese provinces, with additional reports in Japan and South Korea.

SFTSV infections are characterized by a wide spectrum of clinical symptoms [7,8]. The occurrence of shock, respiratory failure, acute respiratory distress syndrome, disseminated intravascular coagulation, central nervous syndrome, or multiple organ failure leads to a higher mortality rate in these patients. Thus, early identification of risk factors associated with the severity of disease would be beneficial [9]. The multiple organ dysfunction stage is characterized by hemorrhagic manifestations, neurologic symptoms, continued decline in platelet numbers, disseminated intravascular coagulation, and multiple organ failure leading to death. Neurological symptoms, which frequently occur in the terminal stage, include lethargy, muscle tremors, convulsions, and coma [7]. Li et al. found that hemorrhagic signs and neurological symptoms were important predictors of fatal outcomes in SFTS [2]. The majority of fatal SFTS cases involved patients over 50 years of age, indicating that advanced age is a risk factor for disease severity and mortality [10]. The importance of viremia evaluation has been reported in several studies, and there is evidence that antiviral therapy has a potential advantage for patients with SFTS [2,11–13].

The examination of a single factor cannot accurately reflect the severity and prognosis of patients with SFTS given the complexity and diversity of its clinical manifestations. Several studies have been conducted with the goal of establishing scoring systems using several specific parameters to predict prognosis in patients with SFTS. Some studies have established models based on laboratory indicators with sensitivities and specificities of 81.5–82.5% and 86.6–91.74%, respectively [14]. Meanwhile, other studies have established models based on clinical symptoms, virological indicators, and laboratory indicators, with sensitivities and specificities of 74.2–77% and 76.1–97%, respectively [15]. Li et al. conducted a large-scale study and established a scoring system comprising of four laboratory variables, age, and neurological symptoms within 6 days after symptom onset. The optimal threshold to predict the risk of death was found to be 8 points (area under the curve [AUC] 0.879, 95% confidence interval 0.855–0.902). However, the clinical scoring system was based on results from a single medical center and did not consider the viral load, an important measure that has previously been reported to correlate strongly with disease outcome [2]. These previous studies were all conducted at a single center or with relatively small sample sizes, and their findings were not validated; furthermore, some studies employed indicators that were not commonly used in the clinical setting, while many hospitals did not investigate relevant indicators. Accordingly, this study aimed to establish a verified model that would be suitable for application in the general population.

No effective vaccine or specific treatment for SFTS is currently available. Treatments for patients with SFTS include ribavirin, gamma globulin administration, steroids, plasma exchange, antibiotics, and symptomatic supportive treatment. To date, no standardized treatment protocol has been established for SFTS. In vitro studies have confirmed the anti-SFTSV effect of ribavirin [3,16,17]. However, clinical studies have demonstrated that ribavirin does not significantly reduce mortality, inhibit the SFTSV viral load, or increase the platelet recovery speed; moreover, it has possible side effects of anemia and hyperamylasemia [18,19]. More than a dozen studies have evaluated the efficacy of ribavirin therapy with conflicting findings due to the lack of multivariate or stratification analyses. Several previous studies demonstrated that gamma globulin had a therapeutic effect on thrombocytopenia [20–22]. However, Shin et al. demonstrated that intravenous gamma globulin was not effective in the treatment of SFTS [23]. Therefore, further studies are required to demonstrate the effectiveness of gamma globulin.

SFTSV infection has been speculated to induce immunosuppression and lead to secondary infection as a result of leukopenia, thrombocytopenia, and interleukin (IL) 10 overexpression [24]. Liu et al. reported that a reduction in $CD3^+$ and $CD4^+$ T lymphocytes in patients with SFTS led to the suppression of immune function, which increases the risk of secondary infection [25]. Speth et al. found that platelet activation resulted in fungal growth inhibition, thereby implying that thrombocytopenia may promote fungal infections [26]. Takeshi et al. reported on a case–control study that demonstrated more frequent secondary infections in the fatal group than in the nonfatal group, including bacterial pneumonia, invasive pulmonary aspergillosis, and bacteremia [27]. The 2010 SFTS guidelines recommended that only patients with severe SFTS should receive antimicrobial treatment[9]. However, which patients should receive prophylactic antibiotics remains unclear. In addition, Shimada et al. used a mouse model to examine the effects of a combination of minocycline and ciprofloxacin on SFTS infection [28]. The results showed that the treatment resulted in prolonged survival times during lethal infection. Therefore, we conducted a subgroup analysis to compare the clinical efficacies of ribavirin, gamma globulin, and antibiotics based on the proposed prediction model.

This study aimed to establish a simple, practical, and stable prognostic model for SFTS using multicenter follow-up data to stratify the criticality of patients with SFTS. Additionally, the study also aimed to evaluate the clinical efficacy and mortality effect of different treatments, including ribavirin, in different subgroups.

## Methods

This study was approved by the Ethics Review Committee (No. 20180034) of Anhui Medical University and conformed to the principles of the Declaration of Helsinki. Written informed consent was obtained from all patients or their immediate relatives. Upon admission, all patients were required to provide informed consent for their clinical data and laboratory test results during hospitalization to be used for clinical research.

This retrospective multicenter study analyzed the data of 377 patients who were diagnosed with SFTS from January 2014 to December 2019. Among them, 161 (modeling group) and 216 (verification group) patients were from the Second Hospital of Anhui Medical University and the First/Provincial Hospital of Anhui Medical University, respectively. More than half of the local cases of SFTS were treated in these hospitals. Blood samples were collected on the day of admission or the next morning.

SFTS was diagnosed based on the criteria of the Guideline for Prevention and Treatment of Severe Fever with Thrombocytopenia Syndrome version 2010 [9]. The diagnostic criteria of SFTSV infection were acute fever with thrombocytopenia and the detection of viral RNA and/or virus-specific immunoglobulin (Ig) M antibody in the peripheral blood. The exclusion criteria were laboratory-confirmed infections with other pathogens, rickets disease, a history of acute or chronic blood disorders, malignant tumors, and acquired immunodeficiency syndrome or other immunodeficiency diseases.

To collect clinical data, we designed a medical record table containing information regarding epidemiology, clinical manifestations, physical examination, laboratory parameters, demographic factors, and date of onset. Two investigators reviewed this information and checked it against the medical records; all data were collected after treatment. The patients who were discharged were followed up for 1 month after discharge.

SFTSV nucleic acid was detected by real-time quantitative reverse transcription-polymerase chain reaction. The primer and probe of the real-time quantitative reverse transcription-polymerase chain reaction test kit were taken from the S fragment of SFTSV. The forward primer was 5'TAAACTTCTGTCTTGCTGGCTCC3', the reverse primer was 5'TGGCAAGATGCCTTCACCA3', and the probe was 5'CGCATCTTCACATTGAT3'. The detection limit was 1000 copies/mL, and 1000 copies/mL were recorded if the test result was negative. Viral load was recorded as copies/mL, and log10 transformed was used for conversion during statistics. SFTSV-specific IgM and IgG antibodies were detected by the gold-standard method, and the kit was purchased from Wuxi Xinlianxin Biomedical Technology Co., Ltd. The first blood sample of SFTS patients was tested for SFTSV by IgM/IgG, and the results were used in this study. The quality control of laboratory data was maintained among the three medical centers, and they underwent quality control inspections by the Anhui Clinical Laboratory Quality Control Center every quarter to ensure data stability. Patients with SFTS were administered ribavirin (0.5 g intravenously bid), antibiotics (cephalosporin or carbapenems, with dosage based on the creatinine clearance rate), or gamma globulin (20 g intravenously qd). Patients who received drug treatment for > 3 or < 3 days were included in the treated and untreated groups, respectively. Moreover, patients with SFTS received systemic support treatment, including the infusion of blood products and granulocyte colony-stimulating factor, and supportive measures, including rest, nutritional support, and electrolyte balance maintenance. Neurologic symptoms were defined as changes in mental state lasting > 24 hours, including lethargy, irritability, or changes in personality and behavior [29]. This study observed that the most common infections of SFTS patients were pneumonia and sepsis. Pneumonia was defined based on the Chinese guidelines for the diagnosis and treatment of hospital-acquired pneumonia and ventilator-associated pneumonia [30]. Diagnosis of sepsis was conducted in accordance

with the guidelines for the diagnosis of sepsis [30] and the diagnosis of pancreatitis according to the diagnostic criteria of acute pancreatitis [31,32]. The infected group included SFTS patients with pneumonia and/or sepsis, while the noninfected group included patients with no signs of infection.

Statistical analyses were conducted using the statistical software package SPSS 22.0 (SPSS, an IBM Company, Armonk, NY, USA). Continuous variables are presented as the mean ± standard deviation or the median and interquartile range. Categorical variables are presented as the frequency or percentage of events. The two-sample *t*-test and Mann–Whitney U test were used for normally and non-normally distributed continuous data, respectively, to determine the relationship between surviving and fatal cases. Some continuous variables were analyzed after transformation to ranked data or logarithmic form. Between-group differences were determined using Pearson's Chi-squared and Fisher's exact tests, as appropriate. Independent risk factors for predicting mortality were identified through univariable and multivariable logistic regression analyses. The regression equation and formula were established through multi-factor logistic regression analysis. The predictive value of the model was evaluated using receiver operating characteristic curve analysis. Further, we determined the cut-off values to yield a simple and reliable model. Statistical graphs were created using GraphPad Prism 8.00 (GraphPad Software, San Diego, CA, USA). $P < 0.05$ was considered statistically significant.

## Results

### Demographic, clinical, and laboratory characteristics

Of 377 patients aged 26 to 86 years, 68 patients died. The average age of included patients was 61.93 ± 10.83 years old. SFTS patients were mainly from the Huaiyang Mountain area. The onset time of SFTS was mainly concentrated between April and November, and the sources of cases in the validation group and modeling group were similar. All patients had flu-like symptoms at the beginning of the illness, such as fever, fatigue, headache, myalgia with other symptoms, and a fever lasting 3–22 days.

There were no significant between-group differences in basic clinical parameters, including sex, age, white blood cell (WBC) count, platelets, and SFTSV RNA, which suggested that the data source of this study was reliable and balanced. However, there were significant between-group differences in some clinical symptoms, including headache, fatigue, and gastrointestinal bleeding (see S1 Table).

### Risk factors for mortality in the modeling group

Of the 161 patients with SFTS in the modeling group, 26 patients (7 men and 19 women) died. Compared with the survival group, the fatal group showed significantly higher incidences of neurologic symptoms (92.31% vs. 6.67%, odds ratio [OR] = 168), gastrointestinal bleeding (76.92% vs. 28.89%), ecchymosis (76.92% vs. 35.56%), and pancreatitis (88.46% vs. 62.22%). Univariate regression analysis revealed significant differences between the survival and fatal groups in age, red blood cell count, hemoglobin, platelets, calcium, blood glucose, estimated glomerular filtration rate, alanine transaminase, aspartate aminotransferase, prothrombin time—international normalized ratio, D-dimer, amylase, human placental lactogen, SFTSV-specific IgM antibodies, SFTSV RNA, pancreatitis, gastrointestinal bleeding, mucosal ecchymosis, neurologic symptoms, and urinary protein (all $P < 0.05$) (Table 1). Based on the univariate analysis, we selected 14 parameters as independent parameters. Multiple regression analysis revealed a strong correlation between neurologic symptoms and fatal outcomes ($P < 0.001$, OR = 142.9). Given that the hazard ratio of neurologic symptoms was 142.9, its

**Table 1. Comparison of Clinical Features of Patients with SFTS in the Survival and Death Subgroups of the Modeling Group (% or Range).**

| Characteristics | Survival Group (n = 135) | Death Group (n = 26) | OR Value | *P* Value |
|---|---|---|---|---|
| **Demographic feature** | | | | |
| Male | 56 (41.48) | 7 (26.92) | 0.520 | 0.164 |
| Age, years | 64 (53, 69) | 67 (60.75, 72) | 1.054 | 0.041 |
| **Clinical manifestation on admission** | | | | |
| Headache | 72 (53.33) | 15 (57.69) | 1.193 | 0.683 |
| Fatigue | 102 (75.56) | 19 (73.08) | 0.878 | 0.789 |
| Myalgia | 90 (66.67) | 16 (61.54) | 0.800 | 0.614 |
| Nausea and Vomiting | 97 (71.85) | 21 (80.77) | 1.645 | 0.347 |
| Lymphadenopathy | 52 (38.52) | 14 (53.85) | 1.862 | 0.146 |
| Pancreatitis | 84 (62.22) | 23 (88.46) | 4.655 | 0.009 |
| Gastrointestinal bleeding | 39 (28.89) | 20 (76.92) | 8.205 | <0.001 |
| Ecchymosis | 48 (35.56) | 20 (76.92) | 6.042 | <0.001 |
| Accompanied infection | 61 (45.19) | 16 (61.54) | 0.515 | 0.126 |
| Neurologic symptoms | 9 (6.67) | 24 (92.31) | 168.000 | <0.001 |
| **Laboratory tests** | | | | |
| SFTS IgM (positive) | 92 (68.15) | 11 (42.31) | 0.343 | 0.012 |
| SFTS IgG (positive) | 21 (15.56) | 0 (0) | 0.000 | 0.066 |
| SFTSV RNA (lg, copies/ml) | 3.66 (3.00, 4.84) | 5.28 (4.06, 6.08) | 2.028 | <0.001 |
| White blood cells ($\times 10^9$/L) | 2.30 (1.45, 3.64) | 1.76 (1.33, 2.23) | 0.901 | 0.085 |
| Neutrophils ($\times 10^9$/L) | 1.23 (0.79, 2.19) | 1.03 (0.64, 1.39) | 0.923 | 0.198 |
| Lymphocytes ($\times 10^9$/L) | 0.59 (0.40, 1.16) | 0.51 (0.40, 0.91) | 0.612 | 0.263 |
| Eosinophil (%) | 0.00 (0.00, 0.10) | 0.00 (0.00, 0.00) | 0.011 | 0.078 |
| Red blood cells ($\times 10^9$/L) | 4.28 (3.99, 4.60) | 3.97 (3.65, 4.42) | 0.425 | 0.021 |
| Hemoglobin (g/L) | 128 (119.5, 139) | 119.5 (109, 132.75) | 0.973 | 0.028 |
| Platelets ($\times 10^9$/L) | 50 (33, 70) | 34 (20, 45) | 0.963 | <0.001 |
| ALT (U/L) | 74 (45, 122) | 109 (58, 189) | 1.003 | 0.029 |
| AST (U/L) | 155 (103, 269) | 446 (200, 711) | 1.001 | <0.001 |
| TBIL (μmol/L) | 9.8 (7.4, 13.2) | 9.1 (6.2, 12.0) | 1.008 | 0.469 |
| eGFR(ml. min/1.73 ml$^2$) | 89.61 (70.66, 113.64) | 71.16 (45.79, 94.35) | 0.979 | 0.003 |
| Glucose (mmol/L) | 6.17 (5.27, 7.46) | 7.62 (5.76, 9.8) | 1.122 | 0.029 |
| Calcium (mmol/L) | 1.93 ± 0.14 | 1.84 ± 0.18 | 0.023 | 0.008 |
| CK (U/L) | 366 (190, 807) | 484 (327, 1754) | 1.000 | 0.069 |
| LDH (U/L) | 563 (363, 828) | 725 (545, 1383) | 1.001 | 0.006 |
| PT-INR | 1.02 (0.97, 1.11) | 1.10 (1.01, 1.25) | 74.745 | 0.012 |
| D-D dimer (ng/L) | 2.53 (1.25, 4.79) | 7.15 (4.58, 10.60) | 1.094 | <0.001 |
| Amylase (U/L) | 113 (69, 178) | 159 (102, 219) | 1.001 | 0.037 |
| Lipase (U/L) | 151 (81, 386) | 278 (151, 524) | 1.000 | 0.044 |
| Urinary occult blood (positive) | 40 (29.63) | 9 (34.62) | 1.257 | 0.613 |
| Urine protein (positive) | 69 (51.11) | 22 (84.62) | 5.261 | 0.024 |
| Interleukin 6 (pg/ml) | 37.9 (17, 65) | 129.25 (61.88, 230.5) | 1.006 | <0.001 |

Ranges of 2 and above were defined as positive, while ranges of 2 and below were defined as negative for urinary occult blood and urine protein. OR: Odds ratio; SFTS: Severe fever with thrombocytopenia syndrome; IgM: Immunoglobulin M; IgG: Immunoglobulin G; SFTSV: SFTS virus; ALT: Alanine aminotransferase; AST: Aspartate transaminase; TBIL: Total bilirubin; eGFR: Glomerular filtration rate; CK: Creatinine kinase; LDH: Lactate dehydrogenase; PT-INR: Prothrombin time-internationalization ratio.

**Table 2. Multivariate Logistic Analysis of Factors Affecting the Prognosis of Patients with SFTS in the Modeling Group.**

| Variable | *B* value | SE | Wald | OR Value (95% CI) | *P* Value |
|---|---|---|---|---|---|
| **Age** | 0.094 | 0.046 | 4.218 | 1.099 (1.004, 1.203) | 0.040 |
| **White blood cells** | 0.119 | 0.109 | 1.185 | 1.126 (0.909, 1.395) | 0.276 |
| **Platelets** | -0.012 | 0.014 | 0.736 | 0.988 (0.960, 1.016) | 0.391 |
| **ALT** | 0.002 | 0.003 | 0.646 | 1.002 (0.997, 1.008) | 0.421 |
| **eGFR** | -0.012 | 0.012 | 1.038 | 0.988 (0.966, 1.011) | 0.308 |
| **CK** | 0.000 | 0.000 | 0.052 | 1.000 (1.000, 1.000) | 0.820 |
| **Gastrointestinal bleeding** | 2.583 | 0.899 | 8.257 | 13.232 (2.273, 77.027) | 0.004 |
| **Pancreatitis** | 1.146 | 0.917 | 1.559 | 3.144 (0.521, 18.984) | 0.212 |
| **Urine protein** | 0.314 | 0.914 | 0.118 | 1.369 (0.228, 8.219) | 0.731 |
| **Hemoglobin** | -0.009 | 0.020 | 0.210 | 0.991 (0.954, 1.030) | 0.647 |
| **SFTS IgM /IgG** | | | 0.476 | | 0.924 |
| **SFTSV RNA** | 0.599 | 0.305 | 3.860 | 1.821 (1.001, 3.310) | 0.049 |
| **Constant** | -15.110 | 5.717 | 6.985 | 0.000 | 0.008 |

OR: Odds ratio; SFTS: Severe fever with thrombocytopenia syndrome; SFTSV: SFTS virus; ALT: Alanine aminotransferase; eGFR: Glomerular filtration rate; CK: Creatinine kinase; Ig: Immunoglobulin.

inclusion into the multivariate analysis may have affected the value of the other variables. Therefore, we selected the remaining 13 variables as independent variables in the binary logistic regression model. Age, SFTSV RNA, and gastrointestinal bleeding were independent risk factors for fatal outcomes in patients with SFTS in the modeling group (Tables 1, 2 and S2).

## Predictive value of the joint indices score and neurologic symptoms in the modeling group

Based on the multivariate analysis, we performed a binary logistic regression analysis using age, SFTSV RNA, and gastrointestinal bleeding; moreover, we established the regression equation formula. Given the convenience of clinical use, we chose the coefficient as an integer and obtained the following formula:

$$\text{The joint indices score} = \text{Age} + 10 \times \text{SFTSV RNA (lg)} + 30 \times \text{Gastrointestinal bleeding} \quad (1)$$

In the equation above, the cut-off value was 118. Receiver operating characteristic curve analysis revealed that the cut-off value of the AUC was 0.859, with a sensitivity and specificity of 0.962 and 0.677, respectively; moreover, the diagnostic index was 0.638. The AUC and diagnostic index of the joint indices score were higher than those of age and SFTSV RNA. The AUC of neurologic symptoms for predicting fatal outcomes in patients with SFTS was 0.931, with a sensitivity and specificity of 0.923 and 0.940, respectively. The diagnostic index was 0.863 (Fig 1 and Table 3).

To evaluate the consistency of the joint indices score and neurologic symptoms, patients with SFTS were divided into four groups based on both indicators. We found that 29 patients tested positive for both indicators, 40 were positive for the joint indices score and negative for neurologic symptoms, 4 were negative for the joint indices score and positive for neurologic symptoms, and 88 were negative for both. Concordance analysis of the joint indices score and neurologic symptoms showed statistically significant differences (P < 0.001). The mortality rates of the positive and negative groups for the joint indices score, including age, SFTSV RNA, and gastrointestinal bleeding, were 36.2% and 1.1%, respectively, while the mortality rates of the positive and negative groups for neurologic symptoms were 72.7% and 5.8%, respectively (Fig 1).

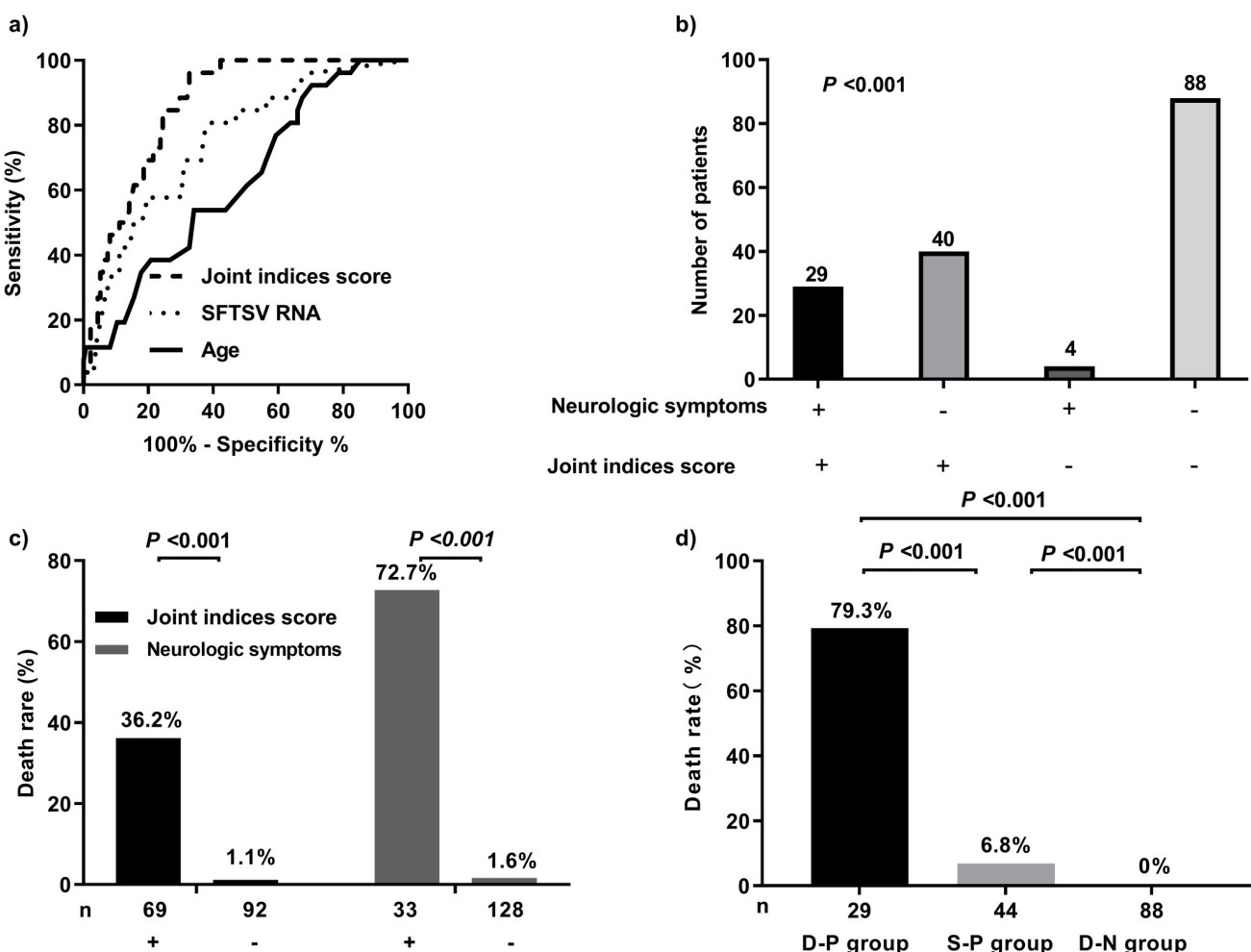

**Fig 1. A clinical scoring model for predicting mortality in patients with severe fever with thrombocytopenia syndrome (SFTS).** (A) Receiver operating characteristic (ROC) curve for predicting mortality in patients with SFTS. (B) Consistency between joint indices score and neurologic symptoms. (C) Mortality distribution of patients with SFTS according to each indicator. (D) Mortality distribution of patients with SFTS according to both combined indicators. Note: Joint indices score = Age + 10 × SFTSV RNA (lg) + 30 × Gastrointestinal bleeding. D-P: double-positive, S-P: single-positive, D-N: double-negative, M-G: modeling group, V-G: verification group.

## Predictive value of joint indices score and neurologic symptoms in the verification group

The verification group comprised 216 patients; among them, 42 patients (18 men and 24 women) died. The inclusion and exclusion criteria were similar to those in the modeling group. The fatal group had a higher median age than the survival group (70 years vs. 61.5 years, P < 0.001). Compared with the survival group, the fatal group had significantly higher incidences of neurologic symptoms (76.19% vs. 6.23%, OR = 47.418) and gastrointestinal bleeding (57.14% vs. 11.49%) (see S3 Table).

The AUC of the joint indices score was 0.897, with a sensitivity and specificity of 0.805 and 0.882, respectively; moreover, the diagnostic index was 0.687. The AUC of neurologic symptoms was 0.858, with a sensitivity and specificity of 0.780 and 0.935, respectively; further, the diagnostic index was 0.715. We found that 33 patients tested positive for both indicators; 44 (27.3%) were positive for the joint indices score and negative for neurologic symptoms, 10 were negative for the joint indices score and positive for neurologic symptoms, and 137 were

**Table 3. Forecast Performance Analysis of Each Index in the Modeling and Verification Groups.**

| Variable | Groups | Cut-off Value | Sensitivity | Specificity | Diagnostic Index | AUC (95% CI) | P Value |
|---|---|---|---|---|---|---|---|
| **Age** | **Modeling group** | **54.50** | **0.923** | **0.301** | **0.224** | **0.632 (0.523, 0.742)** | **0.033** |
| | **Verification group** | | **0.683** | **0.716** | **0.399** | **0.763 (0.691, 0.835)** | **<0.001** |
| **SFTS RNA** | Modeling group | 3.987 | 0.808 | 0.624 | 0.432 | 0.752 (0.654, 0.85) | <0.001 |
| | Verification group | | 0.780 | 0.828 | 0.609 | 0.835 (0.756, 0.915) | <0.001 |
| **Gastrointestinal bleeding** | Modeling group | | 0.769 | 0.707 | 0.476 | 0.738 (0.634, 0.842) | <0.001 |
| | Verification group | | 0.561 | 0.882 | 0.443 | 0.721 (0.624, 0.819) | <0.001 |
| **Joint indices score** | Modeling group | 118.14 | 0.962 | 0.677 | 0.638 | 0.859 (0.799, 0.919) | <0.001 |
| | Verification group | | 0.805 | 0.882 | 0.687 | 0.897 (0.841, 0.954) | <0.001 |
| **Neurologic symptoms** | Modeling group | | 0.923 | 0.940 | 0.863 | 0.931 (0.868, 0.995) | <0.001 |
| | Verification group | | 0.780 | 0.935 | 0.715 | 0.858 (0.780, 0.936) | <0.001 |

Joint indices score = Age+ 10× SFTSV RNA (lg)+30× Gastrointestinal bleeding; AUC: Area under the curve; SFTS: Severe fever with thrombocytopenia syndrome; SFTSV: SFTS virus.

negative for both indicators. Concordance analysis of the joint indices score and neurologic symptoms showed statistically significant differences (P < 0.001) (Table 3 and Fig 2).

## Establishment of a scoring model to predict mortality in patients with SFTS

In the modeling group, the mortality rates of patients in the positive and negative subgroups for the joint indices score were 36.2% and 1.1%, respectively, while the corresponding values

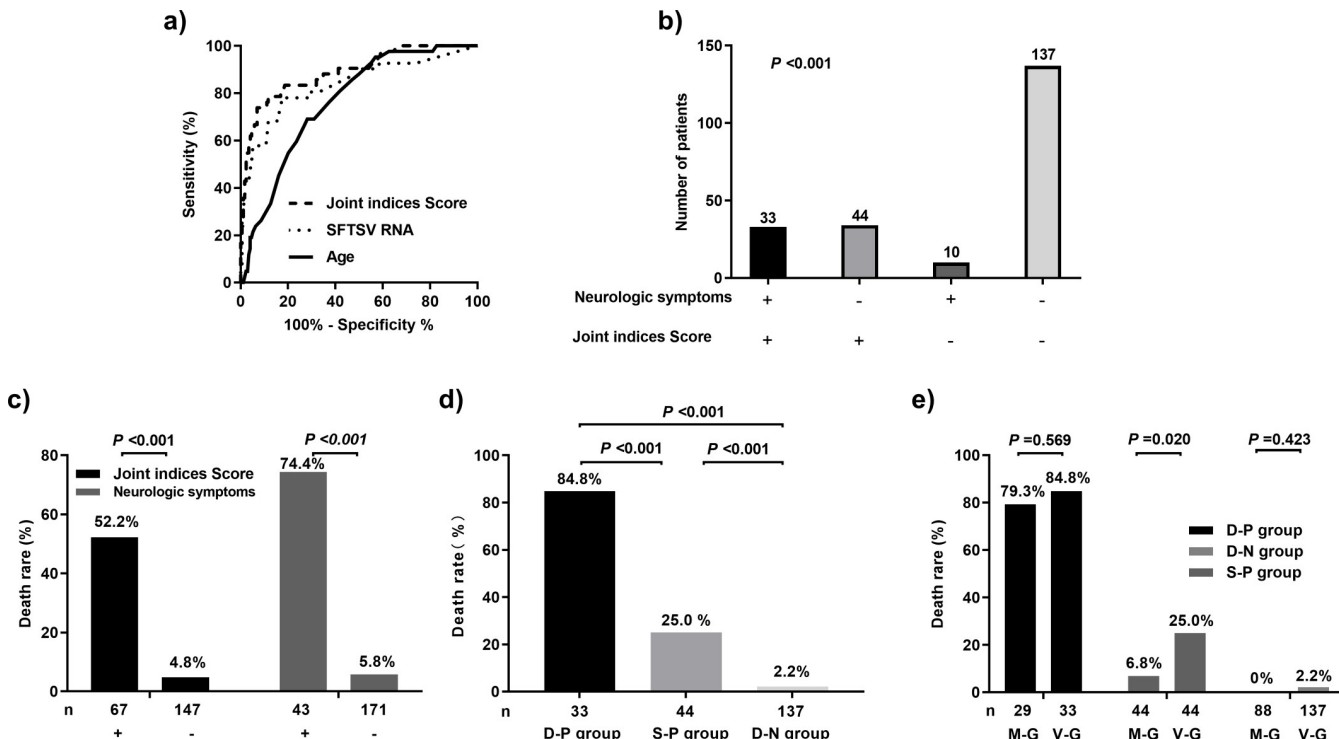

**Fig 2. Verification of the proposed clinical scoring model for predicting mortality in patients with severe fever with thrombocytopenia syndrome (SFTS).** (A) Receiver operating characteristic (ROC) curve for predicting mortality in patients with SFTS in the verification group. (B) Consistency between the joint indices score and the neurological symptoms in the verification group. (C) Mortality distribution of patients with SFTS according to each indicator in the verification group. (D) Mortality distribution of patients with SFTS according to the two combined indicators in the verification group. (E) Comparison of mortality in subgroups in the modeling and verification groups.

for neurologic symptoms were 72.7% and 1.6%, respectively. In the verification group, the mortality rates of the patients in the positive and negative groups for the joint indices score were 52.2% and 4.8%, respectively, while the corresponding values for neurologic symptoms were 74.4% and 5.8%, respectively. Patients who were positive for both indicators, positive for one indicator, or negative for both indicators were included in the double-positive, single-positive, and double-negative subgroups, respectively. In the modeling group, the mortality rates in the double-positive, single-positive, and double-negative subgroups were 79.3%, 6.8%, and 0%, respectively, while the corresponding values in the verification group were 84.8%, 25.0%, and 2.2%, respectively. The combination of both indicators had a higher predictive value than a single indicator; accordingly, we established a novel prognostic prediction model for SFTS by combining the joint indices score and neurologic symptoms (Figs 1 and 2).

There was a significant difference in the mortality rate between the modeling and verification groups in the single-positive subgroup (P < 0.05) but not in the double-positive and double-negative subgroups (both P > 0.05). This indicated consistent distribution of the fatal outcomes in the modeling and verification groups (Table 3 and Fig 2).

## Drug efficacy evaluation based on the model

The analysis of the single-positive subgroup showed that the mortality rate was only 7.4% in the ribavirin-treated group, while it was 29.4% in the non-ribavirin-treated group (P = 0.006). However, there were no significant differences between patients treated with ribavirin and those who were not treated with it in the double-positive and double-negative groups (both P > 0.05). Similarly, there were also no significant differences in the mortality rate between the groups of patients treated and not treated with gamma globulin in the double-positive and double-negative groups (both P > 0.05). Further, gamma globulin treatment did not significantly reduce the mortality rate in the single-positive subgroup (P = 0.09) (Fig 3).

In this study, we performed a subgroup analysis of SFTS patients treated with antibiotics according to the presence or absence of concomitant infection or agranulocytosis. There were 175 patients in the group with concomitant infection or agranulocytosis (152 with concomitant infection and 35 with agranulocytosis), and 202 patients did not have infection or agranulocytosis. Of the patients with concomitant infection, 144 had a pulmonary infection and 18 had a bloodstream infection; the pathogenic bacterial species were *Escherichia coli* in 15 cases of sepsis and *Klebsiella pneumoniae* in the remaining 3 cases. There were 75 patients with a simple bacterial infection, 10 patients with a fungal infection, and 59 patients with a mixed infection. There were 30 patients with a positive sputum culture in the pneumonia group, including 20 cases of *E. coli*, 7 cases of *Aspergillus*, and 3 cases of *Acinetobacter baumannii*; the remaining pneumonia patients were clinically diagnosed.

There was a significant difference in the mortality rate between the patients who were treated with antibiotics and those who were not treated in the overall analysis (P < 0.001). The mortality rate was only 7.2% in the antibiotic-treated group and 47.4% in the non-antibiotic-treated group (P < 0.001) in the single-positive subgroup, even in individuals without significant granulocytopenia and infection, and early prophylaxis was associated with reduced mortality (9.0% vs. 22.8%, P = 0.008). However, there were no significant differences in the mortality rate between the groups of patients treated and not treated with antibiotics in the double-positive and double-negative groups (both P > 0.05) (Figs 3 and 4).

Patients were divided into two groups based on whether they were accompanied by infection or agranulocytosis. Agranulocytosis was defined as a peripheral blood neutrophil level of less than $0.5 \times 10^9$/L. Antibiotic treatment significantly reduced mortality in the single-positive

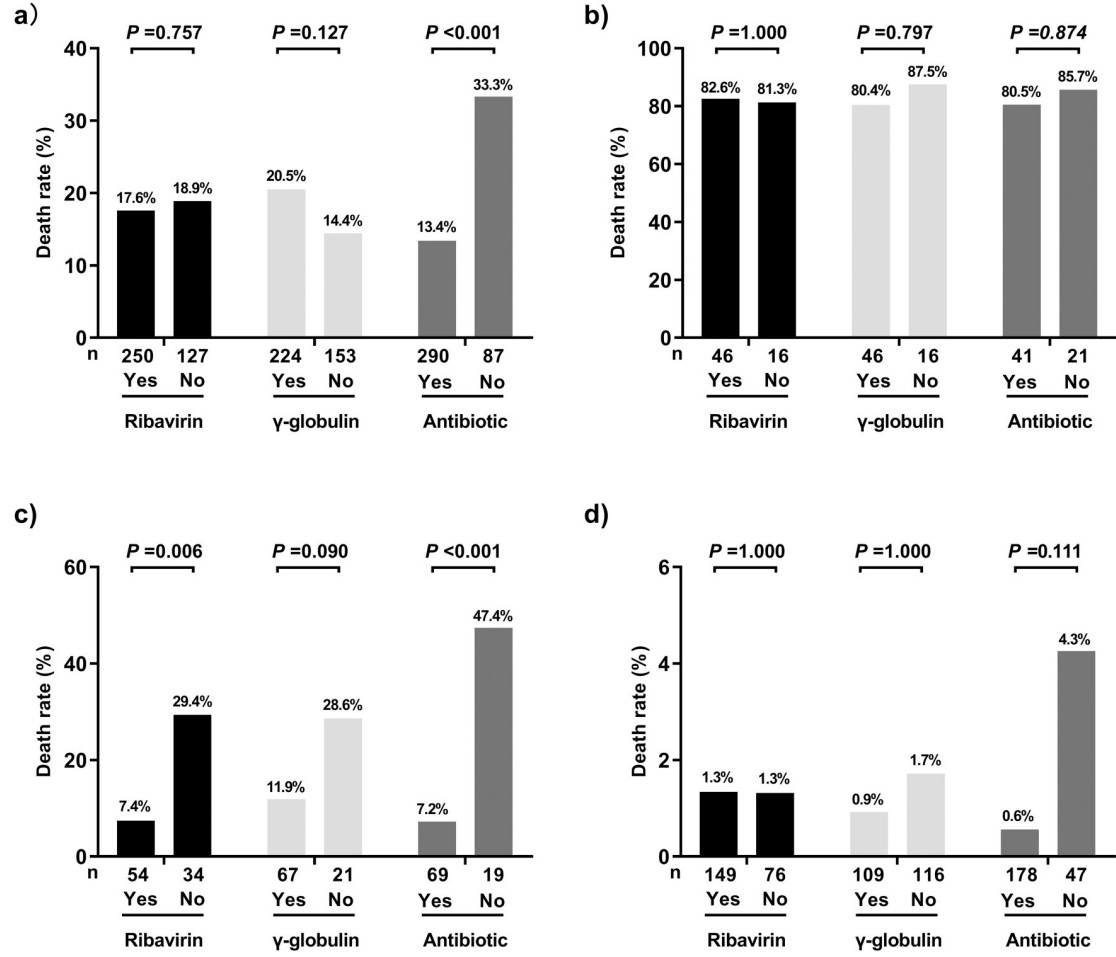

**Fig 3. Analysis of the effect of three drugs on the outcome of patients with severe fever with thrombocytopenia syndrome.** (A) Overall analysis. (B) Double-positive group. (C) Single-positive group. (D) Double-negative group.

group (P = 0.003 in patients with infection group or agranulocytosis, and P = 0.045 in patients with non-infection and agranulocytosis, respectively) but not in the double-positive and double-negative groups (all P > 0.05). Among the 88 patients in the single-positive group who were treated with antibiotics, we assessed the WBC count, percentage of neutrophils (N%), C-reactive protein (CRP) level, and procalcitonin (PCT) level according to the presence of infection and agranulocytosis. There were significant differences in the WBC count and N% (P = 0.014 and P < 0.001, respectively), but not among the other indicators, between the groups of patients with and without agranulocytosis (Fig 4). In addition, all patients with SFTS (N = 377) were divided into an infection group (including sepsis and pneumonia groups) and a non-infection group; the median WBC counts in the sepsis group (n = 17), pneumonia group (n = 142), and non-infection group (n = 223) were $4.620 \times 10^9$/L, $2.035 \times 10^9$/L, and $2.440 \times 10^9$/L, respectively, and the WBC count statistically differed between the pneumonia group and the non-infection group (P = 0.036). The median N% values in the sepsis group (n = 18), pneumonia group (n = 144), and non-infection group (n = 224) were 0.665, 0.620, and 0.630, respectively, and there was no statistical difference between the three groups. The median CRP levels in the sepsis group (n = 16), pneumonia group (n = 134), and non-infection group (n = 214) were 18.80 mg/L, 6.015 mg/L, and 3.920 mg/L, respectively; the CRP level in

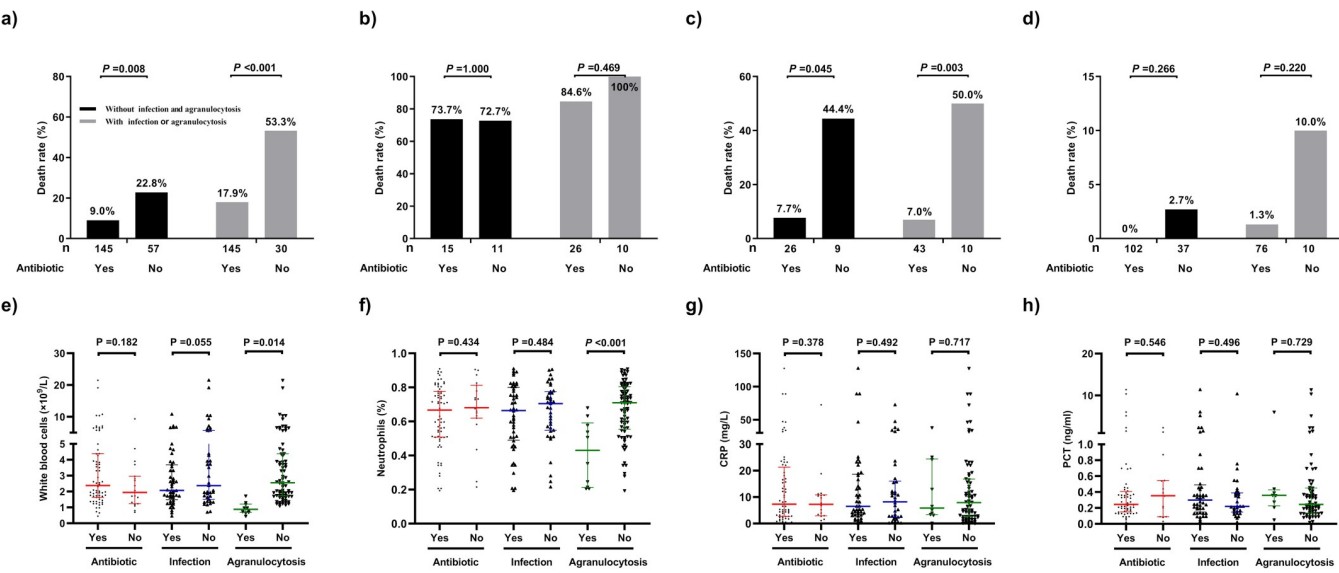

**Fig 4. Subgroup analysis of the antibiotic effect on patients with severe fever with thrombocytopenia syndrome.** (A) Overall analysis. (B) Double-positive group. (C) Single-positive group. (D) Double-negative group and analysis of inflammation indicators. (E) White blood cell (WBC). (F) N%. (G) Procalcitonin (PCT). (H) C-reactive protein (CRP) levels in the single-positive group (N = 88).

the sepsis group was higher than that in the non-infection group (P = 0.009). The median PCT levels in the sepsis group (n = 14), pneumonia group (n = 131), and non-infection group (n = 197) were 0.730 ng/L, 0.244 ng/L, and 0.156 ng/L, respectively, and the PCT levels in the sepsis group and pneumonia group were higher than that in the non-infection group (P = 0.004 and P = 0.026, respectively) (Fig 5).

## Discussion

This study showed that neurologic symptoms, age, SFTSV RNA, and gastrointestinal bleeding were independent risk factors affecting the prognosis of patients with SFTS. Since neurologic symptoms had an overwhelming influence and may have affected the value of other variables, we established a formula based on the remaining three indicators as follows:

$$\text{Joint indices score} = \text{Age} + 10 \times \text{SFTSV RNA (lg)} + 30 \times \text{Gastrointestinal bleeding}$$

The neurologic symptoms and score of the joint indices had good prognostic predictive values; however, they showed poor consistency. Therefore, we combined the score of the joint indices and neurologic symptoms to establish a new predictive model. The mortality rates of the double-positive, single-positive, and double-negative subgroups were 79.3%, 6.8%, and 0%, respectively. The model was validated by data obtained from two other medical centers, which yielded similar results.

Several previous studies have proposed predictive models related to the outcome of SFTS; however, they have yielded inconsistent results [33]. Xiong et al. developed a model (AUC = 0.965) for classifying patients with SFTS using central nervous system symptoms, respiratory symptoms, SFTS viral load, and monocyte ratio [14]. Jia et al. included age, serum urea nitrogen, and activated partial thromboplastin time to construct a mortality prediction model for SFTS, with an AUC of 0.927 [33]. Yang et al. established a new severity scoring system using the state of consciousness scale, lactate dehydrogenase, activated partial thromboplastin time, and oxygen saturation for predicting the prognosis of patients with SFTS [34];

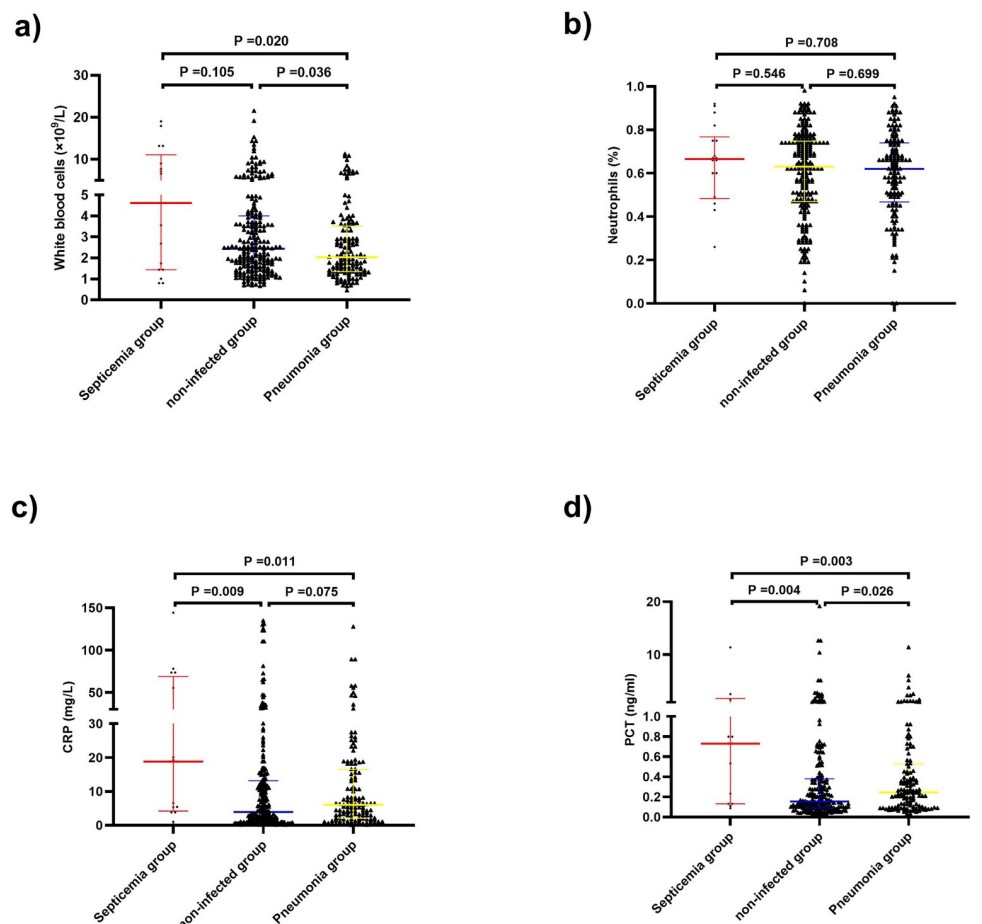

**Fig 5. Analysis of inflammatory parameters of patients with severe fever with thrombocytopenia syndrome.** (A) White blood cell (WBC); (B) N%; (C) C-reactive protein (CRP); (D) Procalcitonin (PCT) levels in all patients with SFTS (N = 377).

further, the sensitivity and specificity of the degree score for mortality risk prediction in hospitalized patients were 74.2% and 76.1%, respectively. Wang et al. reported that age and serum aspartate aminotransferase and creatinine levels were independent risk factors for death. The AUC value for predicting mortality was 0.892, while the sensitivity and specificity were 82.5% and 86.6%, respectively [15]. Contrastingly, the sensitivity and specificity of our model were 25.0% and 84.8%, respectively, in the modeling group and 61.5% and 86.3%, respectively, in the verification group (See S4 Table). However, the previous models were established by single-center studies or employed an overly complicated formula calculation method. Further, several models only included laboratory indicators, which did not account for clinical symptoms, including neurologic symptoms. In our study, neurologic symptoms had an overwhelming impact and showed poor consistency with other indicators (the joint indices score); therefore, they were addressed separately. Furthermore, the multivariate analysis indicated that serum urea nitrogen, prothrombin time, and lactate dehydrogenase were not independent risk factors; therefore, the models containing these indicators were unsuitable. Additionally, we included glomerular filtration rate as a parameter reflective of renal metabolism; however, the multivariate analysis suggested that glomerular filtration rate was not an independent risk factor. Finally, the above model showed poor prediction. Given the aforementioned reasons,

these models cannot be popularized and applied. We found that the combined use of neurologic symptoms and the ARBS had a good prognostic prediction effect for SFTS, which was confirmed in patients from other centers, indicating that the model was reliable and clinically feasible.

Neurologic symptoms are common complications of SFTS. The detection of SFTSV and increased cerebrospinal fluid levels of monocyte chemoattractant protein-1 and IL-8 suggest that neurologic symptoms may be caused by increased cytokine levels in the cerebrospinal fluid [35]. Additionally, Kaneko et al. reported that rapid progression of neurologic symptoms in patients with SFTS might result from damage to the brain tissue caused by SFTSV invasion [36]. Related studies have demonstrated that SFTSV can invade the central nervous system and cause viral encephalitis, with most of its neurologic symptoms involving the cerebral cortex. We observed a strong correlation (OR = 142.9) between neurologic symptoms and mortality in SFTS, which suggests that neurologic symptoms may be a critical risk factor for mortality. Given the inconsistency between the score of the joint indices and neurologic symptoms, as well as the combination of both indicators having a higher predictive value than a single indicator, we established a new model for SFTS based on both indicators.

Several risk factors [37–42] for mortality have been identified in patients with SFTS, including advanced age and SFTSV RNA. Similarly, we found that age was an independent risk factor for mortality in patients with SFTS. Most patients with SFTS were older adults, with a high prevalence of underlying diseases among them, which may have increased their mortality risk. A high viral load has been shown to be a strong risk factor for fatal outcomes in patients with SFTS [43–45], which was consistent with our findings. Cell-free DNA levels (> 711.7 ng/mL) predict severe illness in patients with SFTS [46], which may result in excessive cytokine secretion, thereby aggravating disease progression further. Moreover, we observed an increased mortality rate among SFTS patients with gastrointestinal bleeding. Severe thrombocytopenia in patients with SFTS may induce decreased thrombin synthesis; further, SFTSV increases vascular permeability by damaging vascular endothelial cells. This causes extensive skin ecchymosis, as well as tissue and organ bleeding, in patients with SFTS, resulting in disease aggravation and increased mortality [47,48]. Wang et al [49] suggested that the nucleocapsid protein (NP)-specific IgM-delayed patients tend to have severe clinical presentations compared to NP-specific IgM-positive patients. Specifically, patients with mild disease had significantly higher levels of NP-specific IgM titer on days 7–9 of the disease onset compared to severe and fatal patients (P = 0.016). On days 10–13, substantially higher levels of IgM responses were shown in mild SFTS patients, compared to severe and fatal patients (P = 0.002). In the present study, we showed that the positive rate of SFTS-specific IgM antibody was significantly different between the survival and death groups (68.15% vs 42.3%, P = 0.012), which was consistent with the above results. Similarly, we also detected IgM antibodies against NP fragments within two weeks of onset. Our study demonstrated that the production of early specific IgM antibodies has a protective effect on SFTS patients, possibly because the early specific antibodies may have a certain inhibitory effect on virus replication. Meanwhile, it was speculated that immunosuppression might be the cause of delayed production of IgM antibodies, thus causing the exacerbation of disease in SFTS patients. We found that ribavirin lacked an effect on prognosis in patients with SFTS in the overall analysis. However, ribavirin may have had a therapeutic effect on patients with SFTS in the single-positive group. There could be rapid progression in severe (high-risk) patients; moreover, late antiviral treatment cannot quickly block damage caused by immune activation. These patients may rapidly experience respiratory failure and bleeding, which could result in mortality. Therefore, antiviral therapy could have limited mortality reduction and treatment benefits. Although antiviral treatment has a weak effect on mortality, it may accelerate recovery. For patients in the single-positive group, ribavirin can

effectively prevent virus-induced immune damage, which prevents immune disorders and multiple organ failure and accelerates recovery. The inconsistency between previous reports could be attributed to the lack of stratification of models according to the criticality of patients, which may lead to inaccurate evaluation of drug efficacy. A previous study [22] reported that gamma globulin could block abnormal immune responses by slowing disease progression. However, there is a need to further strengthen this evidence regarding clinical efficacy. In the single-positive subgroup, the mortality rates in patients treated and untreated with gamma globulin were 28.6% and 11.9% (P = 0.090), respectively, with gamma globulin showing a non-significant treatment effect. This should be investigated in future large-scale studies. There could be rapid disease progression in patients with severe SFTS, with the patients experiencing multiple organ failure and gamma globulin having limited therapeutic benefits.

SFTS is an infectious viral disease, and it theoretically does not require antibiotic treatment. Since SFTS is a systemic disease causing overall dysfunction of cellular and humoral immunity [50,51], patients with STSF are susceptible to bacterial and fungal infections. Antibiotics are recommended for the prevention and treatment of severe SFTS [9]. Although antibiotics are commonly used to clinically treat viral infectious diseases, there have been few reports on the efficacy of antibiotics in patients with SFTS. Antibiotics had a significant effect on mortality in patients with SFTS in the single-positive group in this study, but not in the double-positive and double-negative groups. Indicators of inflammation, including PCT, are useful parameters for guiding antibiotic therapy in patients with severe sepsis [52]; however, they did not show specificity for secondary bacterial infections in viral diseases [53]. In this study, although the WBC count differed significantly between the pneumonia group and the non-infection group (P = 0.036), the median WBC counts were very similar ($2.035 \times 10^9$/L vs. $2.440 \times 10^9$/L, respectively); both had different degrees of reduction compared to the normal reference value, and there was no significant difference in N% between the infection and non-infection groups. Although the CRP and PCT levels in the sepsis group were higher than those in the non-infection group, the medians were only slightly higher than the reference values. Therefore, it was difficult to judge whether a patient with SFTS had an infection from inflammatory parameters in the early stages of SFTS. Due to low immune function, especially in patients with neutropenia, the infection site was not obvious, or it was difficult to find, and since there was often no positive pathogenic culture result, fever was frequently the only sign of a serious underlying infection in the early stages of SFTS. Most of the pathogenic epidemiological data of granulocytosis with fever in China are about bloodstream infections, which is consistent with global investigations. Gram-negative bacteria are the main pathogenic bacteria, accounting for more than 50% of the causative agents. Common gram-negative bacteria include *Escherichia coli*, *Klebsiella pneumoniae*, and *Pseudomonas aeruginosa*. In addition, gram-positive bacteria also cause the infections, and the main pathogens include Enterococcus, Streptococcus, *Staphylococcus aureus*, and coagulase-negative Staphylococcus [54]. The pathogen spectrum varies with different infection sites and risk factors. For example, *Escherichia coli*, *Klebsiella pneumoniae*, and coagulase-negative staphylococci were the most common isolates after haploidentical hematopoietic stem cell transplantation [55]. In our study, there were 18 cases of bloodstream infection, including 15 cases of *Escherichia coli* septicemia and 3 cases of *Klebsiella pneumoniae* septicemia, which was consistent with the pathogens in the Chinese guidelines for the clinical application of antibacterial drugs for agranulocytosis with fever (2020 Edition). Therefore, we speculated that the infection source was possibly internal. Of note, the evaluation of a patient's condition based on the model established in this study was helpful for the selection of treatment strategies, with initial empiric therapy with antimicrobials being the preferred treatment after risk stratification assessment, without waiting for microbiological results. Additionally, different treatment regimens may affect the prognosis of patients with SFTS, and the treatment

regimens of different medical centers in this study had some differences; these may partially explain the difference in mortality between the single-positive group and the modeling and validation groups.

This study has some limitations. First, it was a retrospective study and therefore a multicenter prospective study is needed. Second, some of the clinical features, such as neurologic symptoms, were subjective indicators. Quantitative analysis should be conducted based on the Glasgow Coma Scale to facilitate improved and stable judgment of neurologic symptoms. Third, Yoo et al. found that IL-6 and IL-10 levels in patients with an SFTSV infection strongly correlated with outcomes [24]. In this study, IL-6 was detected in part of the modeling group of SFTS patients, and the results showed that the level of IL-6 in the death group was higher than that in the survival group, and there was a statistical difference between the two groups (P < 0.001; OR = 1.006) (Table 1 and S1 Fig). However, other cytokines and T cell subtypes were not routinely detected in this retrospective study.

## Conclusion

This multicenter study established and verified a relatively stable and reliable prognostic model for mortality in patients with SFTS. The model could stratify patients with SFTS and accurately determine drug efficacy. It was shown that ribavirin may have a positive therapeutic effect, and patients with SFTS in the double-positive and single-positive groups were suitable for preventive antiviral treatment. Future studies should also investigate the effect of gamma globulin on patients with SFTS. Lastly, antibiotic use in the early disease stages of SFTS may reduce mortality risk.

## Supporting information

**S1 Table. Comparison of clinical features among patients in the modeling and validation groups (% or range).**
(DOCX)

**S2 Table. Multivariate regression analysis of independent risk factors affecting the prognosis of patients with SFTS in the modeling group.**
(DOCX)

**S3 Table. Comparison of clinical features among patients in the survival and death subgroups of the verification group (% or range).**
(DOCX)

**S4 Table. Sensitivity and specificity of the model established by Wang based on our data (Wang et al).**
(DOCX)

**S1 Fig. Comparison of interleukin 6 between the survival and death groups in the modeling group.**
(TIF)

**S1 Data. Clinical and laboratory data of patients with SFTS.**
(XLSX)

## Acknowledgments

We would like to express special thanks to Fen Huang, Dachen Zhou and Lian Zhu for their assistance with data analysis and statistics.

## Author Contributions

**Conceptualization:** Guomei Xia, Zhenhua Zhang.

**Data curation:** Guomei Xia, Shanshan Sun.

**Formal analysis:** Guomei Xia, Cheng Huang, Jun Li, Zhenhua Zhang.

**Funding acquisition:** Zhenhua Zhang.

**Investigation:** Guomei Xia, Shanshan Sun, Shijun Zhou, Guizhou Zou.

**Methodology:** Guomei Xia, Zhenhua Zhang.

**Project administration:** Guomei Xia, Shijun Zhou, Lei Li, Xu Li, Guizhou Zou.

**Resources:** Lei Li, Xu Li, Zhenhua Zhang.

**Software:** Zhenhua Zhang.

**Supervision:** Lei Li, Xu Li, Zhenhua Zhang.

**Validation:** Guomei Xia, Xu Li.

**Writing – original draft:** Guomei Xia.

**Writing – review & editing:** Guomei Xia, Zhenhua Zhang.

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
