## [Decision Letter · Decision Letter 0]

6 Sep 2022

Dear Prof. Zhang,

Thank you very much for submitting your manuscript "A new model for predicting the outcome and effectiveness of drug therapy in patients with severe fever with thrombocytopenia syndrome: A multicenter Chinese study" for consideration at PLOS Neglected Tropical Diseases. As with all papers reviewed by the journal, your manuscript was reviewed by members of the editorial board and by several independent reviewers. In light of the reviews (below this email), we would like to invite the resubmission of a significantly-revised version that takes into account the reviewers' comments. 

We cannot make any decision about publication until we have seen the revised manuscript and your response to the reviewers' comments. Your revised manuscript is also likely to be sent to reviewers for further evaluation.

Sincerely,

Wen-Ping Guo

Academic Editor

Dennis Bente

Section Editor

Reviewer's Responses to Questions

**Key Review Criteria Required for Acceptance?**

**Methods**

-Are the objectives of the study clearly articulated with a clear testable hypothesis stated?

-Is the study design appropriate to address the stated objectives?

-Is the population clearly described and appropriate for the hypothesis being tested?

-Is the sample size sufficient to ensure adequate power to address the hypothesis being tested?

-Were correct statistical analysis used to support conclusions?

-Are there concerns about ethical or regulatory requirements being met?

Reviewer #1: see the reports

Reviewer #2: -Are the objectives of the study clearly articulated with a clear testable hypothesis stated? : YES

-Is the study design appropriate to address the stated objectives? : YES

-Is the population clearly described and appropriate for the hypothesis being tested? : YES

-Is the sample size sufficient to ensure adequate power to address the hypothesis being tested? : YES

-Were correct statistical analysis used to support conclusions? : YES

-Are there concerns about ethical or regulatory requirements being met? : YES

**Results**

-Does the analysis presented match the analysis plan?

-Are the results clearly and completely presented?

-Are the figures (Tables, Images) of sufficient quality for clarity?

Reviewer #1: see the reports

Reviewer #2: -Does the analysis presented match the analysis plan? : YES

-Are the results clearly and completely presented? : YES

-Are the figures (Tables, Images) of sufficient quality for clarity? : YES

**Conclusions**

-Are the conclusions supported by the data presented?

-Are the limitations of analysis clearly described?

-Do the authors discuss how these data can be helpful to advance our understanding of the topic under study?

-Is public health relevance addressed?

Reviewer #1: see the reports

Reviewer #2: -Are the conclusions supported by the data presented? : YES

-Are the limitations of analysis clearly described? : YES

-Do the authors discuss how these data can be helpful to advance our understanding of the topic under study? : YES

-Is public health relevance addressed? : YES

**Editorial and Data Presentation Modifications?**

Reviewer #1: see the reports

Reviewer #2: (No Response)

**Summary and General Comments**

Reviewer #1: In summary, the manuscript was designed to address an important question, however, the study provided no novel findings on improving clinical practice or refine the clinical diagnosis. the methodology is unclear, the writing could be significantly improved in terms of English language as well as overall cohesiveness and clarity. Finally, there are substantial data presented in the results that are not justified in the introduction nor described or defined in the methods. The manuscript is very sloppy constructed and needs to be edited to correct many typos and errors as well as many grammatical errors. Many sentences are incomplete and very difficult to understand. 

Abstract : Levels differed significantly between the infection group and the non-infection group: what do you mean by infection and non-infection group？Need to be clarified in the abstract.

Introduction

* There is only weak description of the background on the modelling study for the risk assessment of SFTS that cannot well support the hypothesis of the current research.

* It does not make sense to claim “, these were all single-center studies with relatively small sample sizes….”, since the Ref 2, over 2000 patients were used for the modelling, despite of single center study, the model was of highly credibility based on large sample size.

--"Physical examination showed tenderness in the upper abdomen and subxiphoid process, blood and urine amylase and lipase levels were significantly increased,” do you mean increased as compared with which group?

Methods

-- In terms of methodology, it is unclear how the cases were approached and recruited and tested. How do you choose the diagnosis method among RT-PCR, IgM?

--The description of the case patient should be strengthened. Was the first blood sample of the case patient tested for SFTSV by IgM/IgG?

-- Please clarify which gene was used to determine the viral load and how? 

--There is a severe concern that the high discrepancy among multiple medical centers on the treatment regimens, actually, it is suspicious that dugs listed in the paper might be of highly diversified for the administered dose or duration across the study hospitals

Results

--The demographic information of the patients are missing. Are the patients recruited from different hospitals comparable for the clinical data and the treatment? 

-- Measurement of viral loads was described in a vague way, and how it was used in the score estimation was not described, was the copy/ml log transformed or otherwise? 

---Was the age used as a continuous variable? Was the Gastrointestinal bleeding used as Yes (1 score ) and no (0 Score)? If yest 30X0 is still 0

---It seems rather arbitrary to exclude neurologic symptoms as a variable because it has a huge OR, then reused it as a separate variable in the positive counting. It seems the most contribution of the death was due to the presence of neurological, considering the mortality rate of 79.3% versus 6.8% between double positive and single-positive. Then why all the efforts to establish another multiple variable score?

Reviewer #2: - A report reported IL-6 and IL-10 levels, rather than viral load and neutralizing antibody titers, in patients with an SFTSV infection strongly correlated with outcomes (for severe disease with an ultimate outcome of recovery or death). 

Do authors also check cytokine in this study? 

- Authors described “Treatments for patients with SFTS include ribavirin, gamma globulin administration, steroids, plasma exchange, and symptomatic supportive treatment on SFTS infection. In vitro studies have confirmed the anti-SFTSV effect of ribavirin (3, 9, 10). However, clinical studies have demonstrated that ribavirin does not significantly reduce mortality” in introduction. 

But, authors also described below

“Ribavirin administration did not have a significant effect on mortality. However, a subgroup analysis revealed a significant treatment effect in the single-positive group (P = 0.006), but not in the double-positive and double-negative groups (both P > 0.99)”

Could authors more detail describe the effect of Ribavirin in manuscript? 

- Authors described that the mortality rates of patients in the positive and negative subgroups for score of the joint indices were 36.2% and 1.1%, respectively and neurologic symptoms, age, SFTSV RNA, and gastrointestinal bleeding were independent risk factors affecting the prognosis of patients with SFTS. 

Are joint indices not independent risk factors affecting the prognosis of patients with SFTS?

PLOS authors have the option to publish the peer review history of their article (what does this mean?). If published, this will include your full peer review and any attached files.

Reviewer #1: No

Reviewer #2: No
---

## [Decision Letter · Decision Letter 1]

12 Dec 2022

Dear Prof. Zhang,

Thank you very much for submitting your manuscript "A new model for predicting the outcome and effectiveness of drug therapy in patients with severe fever with thrombocytopenia syndrome: A multicenter Chinese study" for consideration at PLOS Neglected Tropical Diseases. As with all papers reviewed by the journal, your manuscript was reviewed by members of the editorial board and by several independent reviewers. In light of the reviews (below this email), we would like to invite the resubmission of a significantly-revised version that takes into account the reviewers' comments. 

We cannot make any decision about publication until we have seen the revised manuscript and your response to the reviewers' comments. Your revised manuscript is also likely to be sent to reviewers for further evaluation.

Sincerely,

Wen-Ping Guo

Academic Editor

Dennis Bente

Section Editor

Reviewer's Responses to Questions

**Key Review Criteria Required for Acceptance?**

**Methods**

-Are the objectives of the study clearly articulated with a clear testable hypothesis stated?

-Is the study design appropriate to address the stated objectives?

-Is the population clearly described and appropriate for the hypothesis being tested?

-Is the sample size sufficient to ensure adequate power to address the hypothesis being tested?

-Were correct statistical analysis used to support conclusions?

-Are there concerns about ethical or regulatory requirements being met?

Reviewer #2: acceptance

**Results**

-Does the analysis presented match the analysis plan?

-Are the results clearly and completely presented?

-Are the figures (Tables, Images) of sufficient quality for clarity?

Reviewer #2: acceptance

**Conclusions**

-Are the conclusions supported by the data presented?

-Are the limitations of analysis clearly described?

-Do the authors discuss how these data can be helpful to advance our understanding of the topic under study?

-Is public health relevance addressed?

Reviewer #2: acceptance

**Editorial and Data Presentation Modifications?**

Reviewer #2: acceptance

**Summary and General Comments**

Reviewer #2: Authors described “increases the risk of secondary infection” in line 118. 

And authors described that "especially in patients with neutropenia, the infection site was not obvious, or it was difficult to find, and since there was often no positive pathogenic culture result, fever was frequently the only sign of a serious underlying infection in the early stages of SFTS" in line 481 to 484. 

Could you mention (or suggest) the infection source? for example external or internal. 

SFTSV-specific IgM antibodies are risk factors for mortality in line 221. 

Could the author interpret this in discussion? 

If authors have some incomplete cytokines data, please put this data although authors mentioned are

limitation. Because cytokines are important risk factor and a reference showed that Tocilizumab therapy for IL-6 increment in a patient with non-fatal severe fever with thrombocytopenia syndrome (see https://doi.org/10.1016/j.ijid.2022.06.058).

Authors described “interleukin (IL) 10 overexpression” in line 116. 

Please put the reference (s) in line 116.

PLOS authors have the option to publish the peer review history of their article (what does this mean?). If published, this will include your full peer review and any attached files.

Reviewer #2: No
---

## [Editor Report · Decision Letter 2]

1 Mar 2023

Dear Prof. Zhang,

Please disregard this duplicate of the acceptance email. An error in the system lead to the decision being rescinded and so we are rectifying this by re-accepting the paper. We apologise for any inconvenience caused.

Kind regards,

Alice Musson

Publishing Editor

Dear Prof. Zhang,

We are pleased to inform you that your manuscript 'A new model for predicting the outcome and effectiveness of drug therapy in patients with severe fever with thrombocytopenia syndrome: A multicenter Chinese study' has been provisionally accepted for publication in PLOS Neglected Tropical Diseases.

Best regards,

Elvina Viennet, PhD

Section Editor

Elvina Viennet

Section Editor

---

## [Editor Report · Acceptance letter]

1 Mar 2023

Dear Prof. Zhang,

We are delighted to inform you that your manuscript, "A new model for predicting the outcome and effectiveness of drug therapy in patients with severe fever with thrombocytopenia syndrome: A multicenter Chinese study," has been formally accepted for publication in PLOS Neglected Tropical Diseases.

Best regards,

Shaden Kamhawi

co-Editor-in-Chief

Paul Brindley

co-Editor-in-Chief
